# Navigating the Road to Immunization Equity: Systematic Review of Challenges in Introducing New Vaccines into Sub-Saharan Africa’s Health Systems

**DOI:** 10.3390/vaccines13030269

**Published:** 2025-03-04

**Authors:** Soulama Fousseni, Patrice Ngangue, Abibata Barro, Sophie Wendkoaghenda Ramde, Luc Thierry Bihina, Marie Nicole Ngoufack, Souleymane Bayoulou, Gbetogo Maxime Kiki, Ouedraogo Salfo

**Affiliations:** 1School of Public Health, Texila American University of Zambia, Lilayi 10101, Zambia; ouesa_96@yahoo.fr; 2Faculty of Nursing, Laval University, Quebec, QC G1V 0A6, Canada; 3Institute for Interdisciplinary Training and Research in Health Sciences and Education, Ouagadougou 01 BP 25, Burkina Faso; fleurbarro03@gmail.com; 4Regional Directorate of Health and Public Hygiene of South-Central, Manga 296, Burkina Faso; zoromram_sophie@yahoo.fr; 5Ministry of Public Health, Yaounde 3038, Cameroon; lucthierrybihina3@gmail.com; 6Challenges Initiative Solutions, Yaounde 8550, Cameroon; mnngoufack@yahoo.fr; 7United Nations Population Fund (UNFPA), Ouagadougou 01 BP 575, Burkina Faso; bayoulou.solo@gmail.com; 8Department of Occupational Therapy, Université du Québec à Trois-Rivières, Québec, QC G9A 1R8, Canada; gbokiz@gmail.com

**Keywords:** new vaccines, immunization challenges, implementation barriers, sub-Saharan Africa, health systems strengthening

## Abstract

**Background/Objectives:** Over the past 50 years, developing new vaccines has been pivotal in responding to emerging and re-emerging diseases globally. However, despite substantial partner support, introducing new vaccines in sub-Saharan Africa remains challenging. This systematic review documents the barriers to new vaccine introduction in sub-Saharan Africa by distinguishing between vaccines integrated into routine immunization programs and those introduced primarily for outbreak response. **Methods:** A comprehensive electronic search was conducted across five databases for articles published in English or French on the challenges of new vaccine introduction in sub-Saharan Africa. Three reviewers screened articles independently based on the titles and abstracts, with full-text assessments conducted for inclusion. Data were analyzed thematically and synthesized narratively. **Results:** A total of 796 articles were retrieved from the five databases. Following the screening, 33 articles were finally retained and included in the review. These articles concerned the introduction of eight new vaccines (malaria vaccine, COVID-19 vaccine, HPV vaccine, Ebola vaccine, cholera vaccine, hepatitis B vaccine, rotavirus vaccine, and typhoid vaccine). The analyses revealed coordination and financing challenges for six vaccines in seventeen countries, acceptability challenges for five vaccines in ten countries, logistical challenges for two vaccines in six countries, and quality service delivery challenges for three vaccines in thirteen countries. **Conclusions:** Addressing the challenges of introducing new vaccines in sub-Saharan Africa requires targeted, evidence-based strategies. Prioritizing political commitment, innovative funding, public education, workforce development, and infrastructure improvements will strengthen immunization systems and enable timely vaccine delivery. Collaborative efforts and a focus on local context can advance equitable health outcomes, safeguard public health, and support global immunization goals.

## 1. Introduction

Vaccination remains one of the most effective strategies for preventing and controlling infectious diseases, offering protection against over 20 potentially life-threatening conditions worldwide [1]. Over the past five decades, vaccination has been estimated to have saved more than 130 million lives, underscoring its critical role in global health [2]. Recognizing its importance, the World Health Assembly has recommended the introduction of new vaccines into national immunization programs, particularly to address emerging diseases of public health concern [3].

Routine immunization programs typically focus on childhood and adolescent vaccinations, such as HPV, rotavirus, hepatitis B, and malaria vaccines, which are systematically integrated into national schedules. In contrast, vaccines such as COVID-19, Ebola, cholera, and typhoid are primarily deployed for outbreak response and do not follow routine administration patterns.

Technical and financial assistance have been pivotal to vaccine introduction for resource-limited settings such as sub-Saharan Africa. Between 2000 and 2022, Gavi, the Vaccine Alliance, invested over USD 11 billion to improve access to vaccines across the African continent [4].

Specific initiatives, including the COVAX mechanism funded by the European Union, have supported the deployment of COVID-19 vaccines in 15 African countries [5]. Similarly, the AMVIRA initiative has facilitated malaria vaccine introduction in 19 African nations [6]. Supporting new vaccine rollout also encompasses research, clinical trials [7], and pilot phase implementation [8]. These efforts have led to measurable successes. In Botswana, for instance, introducing COVID-19 vaccines resulted in more than 50% population coverage [9], while the rollout of cholera vaccines successfully curtailed transmission during an outbreak in Lusaka, Zambia [10]. Despite such achievements, significant challenges have been identified in implementing new vaccines in sub-Saharan Africa [11]. First, many new vaccines require stringent storage and transportation conditions to maintain efficacy, which poses difficulties in keeping the cold chain in remote or resource-constrained regions [12]. Second, affordability and funding sustainability challenges threaten coverage rates, particularly in countries with fragile health systems and economies [13].

Additionally, new vaccine introduction frequently encounters resistance fueled by misinformation, mistrust, or entrenched cultural beliefs [14]. Operational challenges further compound these issues. Evaluations of vaccine introductions have highlighted inadequacies in adverse event monitoring systems and limitations in data infrastructure, impeding the comprehensive assessment of vaccine safety and coverage [15]. Furthermore, new vaccines necessitate additional training for healthcare workers, exacerbating workloads in health systems already strained by human resource shortages [16]. Logistical barriers are another key challenge, particularly in reaching remote and marginalized populations [17]. To better understand these challenges, identify effective solutions, and guide the introduction of new vaccines in sub-Saharan Africa, we systematically reviewed published articles examining barriers to vaccine rollout and their associated mitigation strategies in the region. This review includes both routine and outbreak response vaccine categories.

## 2. Materials and Methods

This systematic review followed the PRISMA 2020 Statement guidelines to ensure rigor and transparency in reporting [18]. A comprehensive search strategy, guided by the PICO approach, was developed to identify relevant quantitative and qualitative studies. Five electronic databases—PubMed, Scopus, CINAHL, Web of Science, and Embase—were queried, yielding 796 articles. The search period spanned January 2019 to December 2024. The detailed PubMed search strategy is provided in Appendix A (see Table A1). The review protocol was prospectively registered on the PROSPERO platform (registration number: 532857; registered on 6 April 2024) to enhance methodological transparency and reduce the potential for bias.

### 2.1. Concept Definition

A vaccine is defined as a biological product designed to stimulate the immune system to generate antigen-specific immunity against a pathogen, thereby preventing the disease it causes [19]. A new vaccine is a vaccine recently introduced into the immunization program of a given country [20].

Acceptability refers to the extent to which a vaccine is perceived as appropriate, beneficial, and desirable by the target populations, typically measured by their willingness to be vaccinated and influenced by factors such as trust, accessibility, and perceived risk [21].

### 2.2. Inclusion/Exclusion Criteria

We restricted our search to articles published in English or French between January 2019 and July 2024, focusing on the challenges of introducing new vaccines in sub-Saharan Africa. Articles were excluded if they were: (i) dissertations or theses, (ii) press publications, (iii) editorials, (iv) editorial reports, (v) supplementary articles, (vi) newsletters, or (vii) studies not accessible in the university library.

### 2.3. Selection of Studies

Articles retrieved were imported into Rayyan software for systematic screening (https://www.rayyan.ai). The selection process followed a stepwise approach based on our research question and inclusion criteria. Initially, articles were screened by title and abstract by the principal investigator and collaborators (SF, WSR, LTB, NMN). Of the 796 articles identified, 11 duplicates and 732 articles not meeting the inclusion criteria were excluded. Fifty-three articles underwent full-text review, and twenty were excluded for the following reasons: insufficiently described or missing methodology (n = 5), lack of specificity to new vaccines (n = 4), not vaccine-related (n = 1), not focused on African countries (n = 2), prepublication articles not validated (n = 2), results based on commentary rather than evidence (n = 5), and data dated beyond the eligibility window (n = 1).

Disagreements on eligibility were resolved through consensus among the reviewers, resulting in the inclusion of 33 studies (see Figure 1). These comprised twenty-six quantitative studies, four qualitative studies, two mixed-methods studies, and one clinical trial.

### 2.4. Data Extraction and Analysis

Search results were uploaded to Rayyan for organization and deduplication. Initial selection by three reviewers (title and abstract) was followed by a full-text review based on the inclusion and exclusion criteria. A fourth reviewer resolved disagreements. Narrative synthesis was employed to thematically group the challenges to introducing new vaccines. Extracted data included author and year of publication, study setting, participants and demographics, study design, vaccine type, study quality, and the main challenges reported.

Challenges were categorized into four domains: acceptability, logistical challenges, service delivery, and coordination/financing.

### 2.5. Quality Assessment of Studies

The quality of the included studies was assessed using the Mixed Methods Appraisal Tool (MMAT), 2018 version. The MMAT was selected for its applicability to reviews involving qualitative, quantitative, and mixed-methods studies [22]. Each article was evaluated against five criteria, with a score of two points for “Yes” responses and zero for “No” or “Unknown” responses. Total scores were converted into percentages, ranging from 60% to 100%. Of the 33 studies, 12 scored 60%, 14 scored 80%, and 7 achieved 100%.

## 3. Results

The results of 33 articles concerned the introduction of eight (8) new vaccines. These vaccines include the malaria vaccine (MV), COVID-19 vaccine, human papillomavirus (HPV) vaccine, Ebola virus vaccine, cholera vaccine, hepatitis B vaccine (HBV), rotavirus vaccine, and typhoid vaccine. The introduction of these vaccines essentially revealed four types of challenges. These challenges were grouped into (1) acceptability, (2) immunization service logistics capacity, (3) quality service delivery, and (4) the challenges of coordinating and financing new vaccines.

### 3.1. Acceptability of New Vaccines

Challenges related to the acceptability of new vaccines were reported in 18 of the 33 studies and involved the introduction of five vaccines against malaria, COVID-19, human papillomavirus (HPV), hepatitis B, and Ebola in ten African countries. Acceptability varied by vaccine type and country, ranging from strong adherence to marked reluctance (Table 1).

High vaccine acceptance rates (>90%) were noted for five vaccines across four countries. Key factors driving acceptance included perceived risks of target diseases, such as the Ebola virus in the Democratic Republic of Congo (DRC) [23], the severity of COVID-19 infection in Nigeria [27], the perceived sensitivity of hepatitis B (*p* < 0.05) in Ghana [24], experiential attitudes toward HPV vaccination (*p* < 0.01) in Nigeria [25], and population confidence in malaria vaccine in Sierra Leone [26].

Conversely, vaccine hesitancy emerged in at least eight countries. For COVID-19 vaccines, reluctance was driven by fear of adverse events following immunization (AEFI) in Ethiopia [32] and the DRC [34], lack of information in Senegal [35] and in Sudan [28], low awareness of vaccination in Ethiopia [30], concerns about safety in the DRC [31], religious considerations in Ghana [36], and perceived susceptibility (*p* < 0.001) in Nigeria [37].

For HPV vaccines, barriers to acceptance included parental education level in Ethiopia [29], infodemics in Senegal [38] and in South Africa [33], and trust in regulatory measures in Uganda [39]. Reluctance toward the Ebola vaccine in Guinea stemmed from stigma [7].

### 3.2. Logistic Challenges

Logistical barriers were reported in seven studies concerning the introduction of malaria and HPV vaccines in six countries (Table 2). Challenges included vaccine availability, cold chain infrastructure gaps, and district and facility-level logistics shortfalls.

For the HPV vaccine, specific barriers were global vaccine shortages, delayed rollout in Senegal [38], vaccine unavailability at designated sites in Nigeria [41] and in public programs in South Africa [33], and inadequate cold chain storage in Uganda [39].

For the malaria vaccine, challenges included vaccine supply shortages and resulting community disinterest in Sierra Leone [26] as well as vaccine stock-outs and inadequate mobile logistics in Ghana [42].

Corrective measures adopted included revising logistics plans to address cold chain gaps in Ghana [40], procuring vaccines through Gavi collaborations in Nigeria [41], and integrating HPV vaccination within school programs in South Africa [33].

### 3.3. Challenges to Quality Service Delivery

Five studies identified immunization service delivery challenges for hepatitis B, malaria, and HPV vaccines across 13 countries (Table 3). Key issues included inadequate vaccination delivery strategies and healthcare workforce barriers.

For hepatitis B vaccination, inadequate provision of services contributed to a 32% non-vaccination rate across 12 countries [43]. Long waiting times and overburdened staff during malaria vaccine delivery were similarly noted in Ghana [40]. Furthermore, facility managers criticized extended malaria vaccination schedules for contributing to dropouts [42]. Ineffective HPV vaccine delivery strategies included difficulty tracking school-aged children transferred to other districts in Senegal [38] and in Uganda [39]. Challenges were exacerbated by workforce capacity deficits, particularly in Ghana for malaria vaccines [42] and in Senegal for HPV vaccines [38]. Recommendations included phased program scaling [40], stakeholder engagement [38], comprehensive training of healthcare providers [39], and sharing best practices to improve service delivery [42].

### 3.4. Challenges Associated with Coordinating and Financing New Vaccines

Fourteen studies reported coordination and financing challenges related to six vaccines—malaria, influenza, HPV, rotavirus, typhoid, and hepatitis B—across 17 African countries (Table 4).

Coordination issues included pilot site selection, eligibility criteria for target populations, and implementation planning. In Ghana, the exclusion of districts during malaria vaccine piloting led to dissatisfaction among health officials [42].

Implementation delays were similarly noted. In Senegal, the HPV vaccine rollout was postponed due to vaccine unavailability, insufficient funding, and an incomplete communication plan [38]. Financial resource mobilization challenges impacted the malaria vaccine rollout in Ghana [40].

Financing barriers centered on the high costs of vaccine introduction relative to the WHO-recommended affordability thresholds, as observed with influenza vaccines in Kenya [44] and typhoid vaccines in Malawi [45].

Partner co-financing mitigated some costs, enabling HPV introduction in Senegal [49] and Zimbabwe [47] and rotavirus vaccine deployment in Niger [46]. However, cost variations persisted and were linked to differing proportions of co-financing [33].

Several countries adopted alternative strategies. Niger prioritized lower-cost rotavirus vaccines [46], Mozambique reduced HPV vaccine dosing regimens [48], and Malawi cut ancillary program costs [45]. Without subsidies, financial burdens fell to health services and vaccine beneficiaries, as documented with hepatitis B vaccines in 12 countries [43], HPV programs in Nigeria [33], HPV vaccine in Nigeria [41], and in Sierra Leone, where only 56% of respondents were willing to pay USD 0.69 for the HPV vaccine [26].

## 4. Discussion

This review provides critical insights into the challenges of introducing new vaccines within sub-Saharan Africa’s national immunization programs. By categorizing these challenges into acceptability, logistical capacity, service delivery quality, and coordination and financing, the study identified cross-cutting barriers that are context-dependent and vaccine-specific. These findings are pivotal for informing policymakers, program managers, and stakeholders to optimize vaccine introduction strategies and strengthen regional health interventions.

Our analysis underscores that vaccine acceptability is variable and context-specific, influenced by vaccine type, perceived risk, and sociocultural factors. High acceptance levels (>90%) were documented for Ebola and hepatitis B vaccines among healthcare workers and target populations in the DRC [23] and Ghana [24], respectively. This acceptance included heightened disease risk perceptions, community trust, and provider advocacy.

Conversely, vaccine hesitancy remains a significant barrier, particularly for COVID-19 and HPV vaccines. For COVID-19 vaccines, fear of adverse events following immunization (AEFI), limited information, and religious concerns contributed to reluctance in Ethiopia [32] and Senegal [35]. Similarly, parental education levels and infodemics drove HPV vaccine hesitancy in Senegal [38] and Uganda [39]. These findings mirror those from Latin America, where vaccine acceptability varied widely by country and target group: 95% among healthcare workers in Mexico but as low as 40% in Peru [50]. Such disparities emphasize the critical role of tailored communication strategies, trust-building interventions, and community engagement to mitigate misinformation and address population-specific barriers. Lessons from successful campaigns, such as Ebola vaccination in the DRC, highlight the importance of integrating risk communication and leveraging trusted local actors.

Logistical challenges emerged as central barriers, particularly the limitations of cold chain infrastructure and vaccine supply shortages. Insufficient cold chain capacity hampered HPV vaccine delivery in Uganda [39] and malaria vaccination in Ghana [42], consistent with observations from Latin America, where gaps in cold chain availability similarly undermined the vaccine introduction efforts [51]. In addition to infrastructure gaps, recurrent vaccine shortages, notably for HPV [38] and malaria vaccines [26], disrupted program continuity and undermined population confidence. Comparable vaccine shortages in Asia and the Pacific highlight the global nature of organizational and funding constraints for vaccine distribution [52]. Cold chain strengthening and innovative logistical solutions, such as mobile storage units, must be prioritized to ensure vaccine potency and access in remote settings. Global partnerships, including Gavi and WHO support, remain critical to addressing supply disruptions and bridging logistical inequities across resource-limited contexts.

Shortcomings in immunization service delivery were widespread and included insufficient workforce capacity, poor customer care, and organizational inefficiencies. Long waiting times and provider negligence were reported during malaria vaccine delivery in Ghana [40] and hepatitis B vaccination across 12 countries [43]. Such findings align with studies on COVID-19 vaccination services in Egypt, where site overcrowding and disorganization were predominant challenges [53].

These barriers underscore the need to streamline service delivery processes, optimize workforce distribution, and strengthen training programs. Strategies to address inefficiencies include phased vaccine rollout approaches, stakeholder involvement, and monitoring frameworks to evaluate provider performance and client satisfaction.

Financing challenges remain a cornerstone barrier to vaccine introduction across Africa, where vaccine and operational costs often surpass the available health budgets. For instance, the typhoid vaccine in Malawi [45] and influenza vaccines in Kenya [44] exceeded the WHO-defined cost thresholds for health interventions. Unlike resource-constrained regions in Asia, such as Afghanistan, where HPV vaccination costs represented just 0.7% of GDP due to streamlined target cohorts [54], African countries contend with broader target populations and weaker financing mechanisms.

Co-financing agreements, such as Gavi-supported HPV vaccine programs in Senegal [38] and Zimbabwe [47], offer critical opportunities to offset costs. However, sustained government commitments, innovative financing strategies, and cost-effective program adaptations (e.g., reducing HPV vaccine doses [48]) are imperative to overcome long-term financial barriers and ensure equitable vaccine access.

An important example of a successfully introduced routine childhood vaccine is the rotavirus vaccine, which has now been implemented in 123 countries worldwide, achieving a global coverage of 55% [55]. In sub-Saharan Africa, 38 countries introduced the vaccine between 2012 and 2023, with coverage ranging from 5% to 61% depending on the country [56]. The vaccine is administered orally, concomitantly with other routine childhood immunizations (Penta, PCV, IPV) at 2–4 months of age, depending on the national immunization schedule [57].

Despite its broad introduction, our systematic review did not identify any significant challenges associated with the rotavirus vaccine rollout in the past five years. This may indicate that previous logistical and financial barriers have been successfully mitigated, potentially due to Gavi funding, well-established cold chain infrastructure, and its integration with existing immunization programs. However, the variation in coverage rates (5% to 61%) suggests that access disparities persist, possibly due to regional differences in supply chain efficiency, health system capacity, and caregiver awareness [56].

While our findings did not highlight major recent barriers, the historical challenges faced during the rotavirus vaccine introduction, such as high initial costs, supply chain gaps, and cold chain requirements, remain relevant for countries introducing new routine vaccines in the future. Additionally, continuous monitoring of the coverage rates and assessment of drop-out rates between doses are essential to ensure sustained vaccine impact. Future research should examine whether these challenges persist in specific sub-national or remote regions where routine vaccine distribution remains uneven [58].

This review is a comprehensive repository of challenges encountered in recent vaccine introductions across the African region. It highlights opportunities for strengthening health systems, aligning resources, and tailoring program interventions to overcome contextual barriers.

By addressing these interrelated challenges, countries can improve the efficiency, equity, and success of future vaccine introduction efforts, particularly in emerging diseases and global immunization goals.

### Strengths and Limitations

This review offers a critical overview of vaccine introduction challenges across diverse African settings, providing practical insights to guide immunization program planning and execution. However, limitations must be acknowledged. First, while significant data were synthesized, insufficient evidence on political commitment—an essential determinant of vaccine program success—was available. Second, the exclusion of meta-analytical methods limited the quantitative exploration of results.

Third, a key limitation of this study was the reliance on published scientific literature, which may not have fully captured the breadth of challenges associated with introducing new vaccines in sub-Saharan Africa. Many practical barriers and contextual insights related to vaccine introduction are often documented in gray literature, policy reports, program evaluations, and internal documents from ministries of health, non-governmental organizations (NGOs), and global health agencies such as Gavi, WHO, and UNICEF. By focusing exclusively on peer-reviewed studies in indexed databases, our analysis may have underrepresented operational, logistical, and policy implementation challenges frequently discussed in non-scientific sources. Moreover, publishing implementation experiences and field-based challenges in academic journals can be slow and selective, favoring studies with clear methodologies and measurable outcomes. This poses a risk of publication bias, as critical field-level experiences, such as stakeholder negotiations, real-time decision-making, or adaptive strategies, may not be formally documented in ways that fit within the structured formats of scientific literature. As a result, our findings may skew toward well-documented challenges while underemphasizing nuanced, on-the-ground complexities faced by health workers, policymakers, and community organizations.

Finally, contextual heterogeneity across countries warrants cautious interpretation when comparing findings. Variations in socio-economic, cultural, and health system factors may differentially influence vaccine-related challenges and solutions.

## 5. Conclusions

The successful introduction of new vaccines in sub-Saharan Africa requires context-specific strategies that address both routine immunization challenges (e.g., HPV, rotavirus, hepatitis B, malaria) and emergency response barriers (e.g., COVID-19, Ebola, cholera, typhoid). By distinguishing between these two categories, this review provides a more nuanced understanding of the vaccine implementation hurdles. Strengthening political commitment, financing mechanisms, supply chain logistics, and vaccine acceptance strategies will be essential for future immunization efforts.

Future research should integrate policy reviews, field evaluations, and stakeholder interviews to capture a broader range of implementation experiences, ensuring that both scientific evidence and real-world operational insights inform vaccine introduction policies in sub-Saharan Africa.

## Figures and Tables

**Figure 1 vaccines-13-00269-f001:**
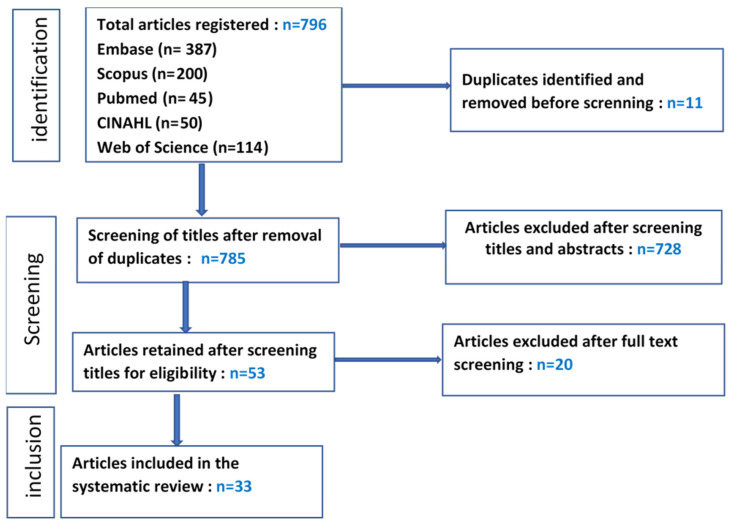
PRISMA flow diagram (PRISMA 2020 flow diagram for new systematic reviews which included searches of databases and registers only).

**Table 1 vaccines-13-00269-t001:** Results on the acceptability of new vaccines.

Author, Year	Study Type	Country	Vaccine	Integrated into Routine Immunization	Participants	Acceptability
**Articles describing high acceptability of new vaccines (adherence > 80%)**						
Doshi et al., 2023 [23]	Quantitative	DRC	Ebola vaccine	No	588 healthcare professionals	99.0% self-reported vaccine acceptance and 70.2% acceptance of the first offer.
Tagbor et al., 2023 [24]	Quantitative	Ghana	Hepatitis B vaccine (HBV)	No	190 nurses	98.8% were willing to receive the vaccine.
Balogun et al., 2022 [25]	Quantitative	Nigeria	HPV vaccine	No	678 parents of teenagers	96.8% of parents intended to have their teenagers vaccinated with HPV.
McCoy et al., 2021 [26]	Mixed study	Sierra Leone	Malaria vaccine	No	615 inhabitants of Bo	95% of participants were ready to receive the vaccine, and 99% were ready to have their children vaccinated.
Azees et al., 2024 [27]	Quantitative	Nigeria	COVID-19 vaccine	No	2130 health workers	92.8% of agents recommend the vaccine despite MAPI.
Elbadawi et al., 2022 [28]	Quantitative	Sudan	COVID-19 vaccine	No	930 healthcare professionals	88.0% of participants agreed to be vaccinated.
**Articles describing reluctance and refusal toward new vaccines**						
Chekol et al., 2022 [29]	Quantitative	Ethiopia	HPV vaccine	Yes	366 schoolgirls	75.7% believe they will take the vaccine if they feel at risk.
Niguse et al., 2023 [30]	Quantitative	Ethiopia	COVID-19 vaccine	No	403 healthcare professionals	71% of participants vaccinated against COVID-19 at least once.
Grantz et al., 2019 [7]	Qualitative	Guinea	Ebola vaccine	No	110 front-line health workers	67% agreed to take part in the vaccine trial.
Barrall et al., 2022 [31]	Quantitative	DRC	COVID-19 vaccine	No	588 healthcare professionals	52.0% reluctant to vaccinate.
Ayele et al., 2024 [32]	Quantitative	Ethiopia	COVID-19 vaccine	No	422 healthcare professionals	45.3% of healthcare professionals accepted the COVID-19 vaccine.
Islam et al., 2021 [33]	Quantitative	South Africa and 4 countries	HPV vaccine	Yes	151 vaccine suppliers	13% of service providers report non-completion of doses (2nd and 3rd).
Garbern et al., 2023 [34]	Quantitative	DRC	COVID-19 vaccine	No	631 healthcare professionals and community members	26.5% of healthcare professionals are reluctant to vaccinate, and 32.4% refuse outright.
Ridde et al., 2021 [35]	Quantitative	Senegal	COVID-19 vaccine	No	607 adults	18.4% do not wish to be vaccinated, 41.5% have not been vaccinated, and 25% have been vaccinated incorrectly.
Addo et al., 2024 [36]	Quantitative	Ghana	COVID-19 vaccine	No	1768 Ghanaian adults	12.7% say their religion does not allow vaccination.
Chinawa et al., 2021 [37]	Quantitative	Nigeria	COVID-19 vaccine	No	577 mothers and children	6.9% of mothers intend to receive the vaccine.
Casey et al., 2022 [38]	Qualitative	Senegal	HPV vaccine	Yes	10 stakeholders	Reluctance and refusal to take the HPV vaccine, according to key informants.
Rujumba et al., 2021 [39]	Qualitative	Uganda	HPV vaccine	Yes	40 health workers, teachers, girls’ parents	Reluctance on the part of some parents (according to teachers).

**Table 2 vaccines-13-00269-t002:** Results of studies on the logistical challenges of new vaccines.

Author, Year	Study Type	Country	Vaccine	Integrated into Routine Immunization	Targets	Logistics Description
Islam et al., 2021 [33]	Quantitative	South Africa	HPV vaccine	Yes	151 service providers	Vaccine not available to the public.
Adjei et al., 2023 [40]	Mixed study	Ghana	Malaria vaccine	Yes	54 facilities and 94 carers	Out of stock in the previous six months in 24% (13/54) HF.
McCoy et al., 2021 [26]	Mixed study	Sierra Leonne	Malaria vaccine	No	615 inhabitants of Bo	Supply shortages during vaccination campaigns.
Nguyen et al., 2020 [41]	Quantitative	Nigeria	HPV vaccine	Yes	137 health agents	Barriers to vaccination: availability of vaccine (39%), lack of CDF (4%), support (1%).
Casey et al., 2022 [38]	Qualitative	Senegal	HPV vaccine	Yes	10 of stakeholders	Vaccine shortages leading to postponements and restrictions on targets.
Rujumba et al., 2021 [39]	Qualitative	Uganda	HPV vaccine	Yes	40 keys informants	Lack of refrigerators in some.
Grant et al., 2022 [42]	Qualitative	Ghana	Malaria vaccine	Yes	21 healthcare managers and staff	Insufficient refrigerators and vaccine carriers in HF.
						Lack of vehicles in the districts and motorbikes in the HF.

**Table 3 vaccines-13-00269-t003:** Results of challenges related to quality service delivery of new vaccines.

Author, Year	Study Type	Country	Vaccine	Integrated into Routine Immunization	Targets	Delivery Challenge
Shah et al., 2020 [43]	Quantitative	12 African countries	HBV	No	1044 healthcare personnel	Customer reception problems 32% of reasons for not vaccinating targets.
Adjei et al., 2023 [40]	Mixed study	Ghana	Malaria vaccine	Yes	54 ESS, 94 carers	Customer reception problems: busy carers (50/54), long waiting times (44/54).
Casey et al., 2022 [38]	Qualitative	Senegal	HPV vaccine	Yes	10 of stakeholders	Unsuitable strategy (focused solely on schools).
						Shortage of qualified staff: lack of information and consistency of services.
Rujumba et al., 2021 [39]	Qualitative	Uganda	HPV vaccine	Yes	40 health workers, teachers, girls’ parents	Insufficient vaccination staff.
						Lack of strategies for those not attending school.
						Non-payment of staff allowances and teacher motivation.
Grant et al., 2022 [42]	Qualitative	Ghana	Malaria vaccine	Yes	21 SSE managers	Lack of training for volunteers responsible for raising community awareness.
						Unsuitable timetable: gap (15 months) between the 3rd and 4th doses, leading to dropouts.

**Table 4 vaccines-13-00269-t004:** Results related to the coordination and financial challenges of new vaccines.

Author, Year	Study Type	Country	Vaccine	Integrated into Routine Immunization	Targets	Coordination and Financial Challenges
Waterlow et al., 2023 [44]	Quantitative	Kenya	Flu vaccines	No	Patients hospitalized in 5 hospitals	Median cost per DALY averted above WHO threshold (USD 100).
Debellut et al., 2022 [45]	Quantitative	Malawi	Typhoid vaccine	Yes	4 districts and 6 establishments	High cost of introduction (4% of total EPI budget, i.e., $29,814,969).
Shah et al., 2020 [43]	Quantitative	12 African countries	HBV	No	1044 healthcare staff	High cost of the vaccine, the most frequently cited reason for non-vaccination (46% of participants).
Islam et al., 2021 [33]	Quantitative	South Africa and 4 countries	HPV vaccine	Yes	151 providers authorized to administer vaccines	High cost of the vaccine for customers is a barrier to vaccination for 30% of interviewers, including 16% of South Africans.
Baral et al., 2021 [8]	Quantitative	Ghana, Kenya and Malawi	Malaria vaccine	Yes	Nourishing SSEs in pilot areas	Financial cost per FVC multiplied by 3 if the government pays for the vaccine in full. Economic costs 3 and 5 times higher.
Debellut et al., 2022 [46]	Quantitative	Niger	Rotavirus Vaccine	Yes	391 children under the age of 5	High cost of the program, varying according to the vaccine: USD 46.7 million with ROTAVAC, USD 61.8 million, ROTASIIL Government co-funding.
Adjei et al., 2023 [40]	Mixed study	Ghana	Malaria vaccine	Yes	54 establishments and 94 carers	Lack of funds forced 90% of establishments to cancel activities.
McCoy et al., 2021 [26]	Mixed study	Sierra Leone	Malaria vaccine	No	615 inhabitants of Bo	Cost of barrier vaccine: 56% prepared to pay USD 0.69 for the vaccine.
Hidle et al., 2022 [47]	Quantitative	Zimbabwe	HPV vaccine	Yes	30 districts and 60 health facilities	MOHCC co-funding 77% high additional cost: USD 7.79 economic cost per dose.
Nguyen et al., 2020 [41]	Quantitative	Nigeria	HPV vaccine	Yes	137 health agents	The cost of vaccine was seen as an obstacle to vaccination by 13% of respondents.
Casey et al., 2022 [38]	Qualitative	Senegal	HPV vaccine	Yes	10 of stakeholders	Delay in the availability of national funds Introduction delayed until Gavi support Communication plan not finalized.
Alonso et al., 2019 [48]	Quantitative	Mozambique	HPV vaccine	Yes	13 keys informants	The budget constraint for 3 doses: economic cost per FIG, 3 doses (USD 2.29) and alternative cost at a reduced dose (USD 31.14).
Grant et al., 2022 [42]	Qualitative	Ghana	Malaria vaccine	Yes	21 health service managers	The pilot district selection process is unknown or contested.
						Criteria for eligibility of targets for vaccination open to criticism.
Brennan et al., 2022 [49]	Quantitative	Senegal	HPV vaccine	Yes	77 health establishments	High operational costs: service provision (57%; USD 4.28 per dose), training (18%; USD 1.36 per dose).

## Data Availability

The original contributions presented in the study are included in this article; further inquiries can be directed to the corresponding author.

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
