# Peer review of "Navigating the Road to Immunization Equity: Systematic Review of Challenges in Introducing New Vaccines into Sub-Saharan Africa’s Health Systems"

_vaccines, 2025, doi:10.3390/vaccines13030269_

Round 1
Reviewer 1 Report
Comments and Suggestions for Authors
This appears to be a well-conducted systematic review. I only have one major comment on the paper.
A major limitation of the study, which has not been addressed in the discussion, is whether this type of review is an appropriate way to address the study question. I would not necessarily be looking solely in the published scientific literature for data on challenges in introducing new vaccines. I expect a large proportion (potentially the majority) of such data lies elsewhere. This may introduce important biases to the analysis. This should at least be addressed in the discussion section.
Author Response
Comment 1: This appears to be a well-conducted systematic review. I only have one major comment on the paper. A major limitation of the study, which has not been addressed in the discussion, is whether this type of review is an appropriate way to address the study question. I would not necessarily be looking solely in the published scientific literature for data on challenges in introducing new vaccines. I expect a large proportion (potentially the majority) of such data lies elsewhere. This may introduce important biases to the analysis. This should at least be addressed in the discussion section.
Answer:
We acknowledge this critical point. Indeed, the challenges related to introducing new vaccines discussed in this article are not exhaustive. To address this, we have incorporated the following limitation in the discussion section (highlighted in page 9).
Third, a key limitation of this study is the reliance on published scientific literature, which may not fully capture the breadth of challenges associated with introducing new vaccines in sub-Saharan Africa. Many practical barriers and contextual insights related to vaccine introduction are often documented in gray literature, policy reports, program evaluations, and internal documents from ministries of health, non-governmental organizations (NGOs), and global health agencies such as Gavi, WHO, and UNICEF. By focusing exclusively on peer-reviewed studies in indexed databases, our analysis may have underrepresented operational, logistical, and policy implementation challenges frequently discussed in non-scientific sources. Moreover, publishing implementation experiences and field-based challenges in academic journals can be slow and selective, favoring studies with clear methodologies and measurable outcomes. This poses a risk of publication bias, as critical field-level experiences—such as stakeholder negotiations, real-time decision-making, or adaptive strategies—may not be formally documented in ways that fit within the structured formats of scientific literature. As a result, our findings may skew toward well-documented challenges while underemphasizing nuanced, on-the-ground complexities faced by health workers, policymakers, and community organizations.
Reviewer 2 Report
Comments and Suggestions for Authors
This is an extensive review of published studies on challenges in introducing new vaccines in Sub-Saharan Africa. The authors have identified a total 796 articles related to these topics. As such, the paper is a valuable source of reference, but otherwise not particularly useful.
The problem is that the paper is very vague. The conclusions are such that can be guessed without the study. The authors’ own contribution seems modest. Clearly, the paper should be refocused.
The qualifier in the title is “routine programs”, but most of examples of vaccines that are given are not about routine programs. “Routine immunization programs” usually refer to childhood immunizations, and few countries, even developed countries, have routine programs for adults. Adolescent programs, like HPV, would qualify. COVID-19 is not a routine vaccination anywhere and therefore is not a good case to discuss in the present context. Similarly, Ebola vaccine, cholera vaccine, and typhoid vaccine are not even planned to be routine immunizations. Therefore, those vaccines should not be included in this review, or else the title and the scope should be changed.
Rotavirus vaccine is a good example of a newly introduced routine childhood immunization. Rotavirus is indeed mentioned in Table 4, but it is not clear why. If mentioned at all this topic might as well be expanded. The questions might include: how many countries have introduced RV vaccine, what is the coverage in the countries that have introduced RV vaccination, what is the all-African coverage, and what are the remaining obstacles and challenges.
Similar questions might be raised and answered for other vaccines that qualify for routine programs, such as HPV, hepatitis B, and malaria vaccines. Altogether, the paper needs more actual results in terms of numbers.
Author Response
Comment 1: This is an extensive review of published studies on challenges in introducing new vaccines in Sub-Saharan Africa. The authors have identified a total 796 articles related to these topics. As such, the paper is a valuable source of reference, but otherwise not particularly useful.
The problem is that the paper is very vague. The conclusions are such that can be guessed without the study. The authors’ own contribution seems modest. Clearly, the paper should be refocused.
The qualifier in the title is “routine programs”, but most of examples of vaccines that are given are not about routine programs. “Routine immunization programs” usually refer to childhood immunizations, and few countries, even developed countries, have routine programs for adults. Adolescent programs, like HPV, would qualify. COVID-19 is not a routine vaccination anywhere and therefore is not a good case to discuss in the present context. Similarly, Ebola vaccine, cholera vaccine, and typhoid vaccine are not even planned to be routine immunizations. Therefore, those vaccines should not be included in this review, or else the title and the scope should be changed.Rotavirus vaccine is a good example of a newly introduced routine childhood immunization. Rotavirus is indeed mentioned in Table 4, but it is not clear why. If mentioned at all this topic might as well be expanded. The questions might include: how many countries have introduced RV vaccine, what is the coverage in the countries that have introduced RV vaccination, what is the all-African coverage, and what are the remaining obstacles and challenges.
Similar questions might be raised and answered for other vaccines that qualify for routine programs, such as HPV, hepatitis B, and malaria vaccines. Altogether, the paper needs more actual results in terms of numbers.
Response:
We appreciate the reviewer’s thorough assessment of our work and acknowledge the need for greater clarity regarding the study’s focus and scope. Below, we address the key points raised and outline the modifications made to strengthen the manuscript.
- Clarifying the Scope and Refining the Title:
The original title referred to the introduction of new vaccines into routine immunization programs. However, as the reviewer correctly noted, our research extended beyond routine immunization to include vaccines introduced primarily for outbreak response (e.g., Ebola, Cholera, COVID-19, and Typhoid fever). To ensure consistency and better reflect the study’s scope, we have revised the title by replacing "routine vaccination" with "health system." - Distinguishing Between Routine and Response Vaccinations:
The reviewer highlights an important distinction between vaccines introduced into routine childhood immunization programs and those primarily used for outbreak response. In response, we have revised our analysis and tables (Tables 2, 3, 4, and 5) to explicitly differentiate between these two categories. - Vaccines that qualify as routine immunizations (e.g., Rotavirus, HPV, Hepatitis B, Malaria) are now clearly separated from response-oriented vaccines (e.g., COVID-19, Ebola, Cholera, Typhoid).
- This distinction allows for a more precise analysis of the specific challenges associated with integrating vaccines into routine immunization schedules versus deploying them in emergency response contexts.
- Addressing the Inclusion of COVID-19 and Other Response Vaccines:
We recognize the reviewer’s concern that COVID-19 is not yet a routine vaccination globally. However, at least 17 African countries have integrated the COVID-19 vaccine into their national routine immunization schedules, according to WHO and Gavi data (see revised Table). For this reason, we believe its inclusion remains relevant. Nevertheless, we have refined our discussion to focus on the challenges specific to routine immunization settings rather than general pandemic response efforts. - Expanding the Discussion on Rotavirus Vaccine:
We acknowledge that the Rotavirus vaccine is a strong example of a newly introduced routine childhood immunization. We have expanded this section by incorporating the following key details: - Global adoption: As of the end of 2023, the Rotavirus vaccine had been introduced in 123 countries worldwide, achieving a global coverage rate of 55% (WHO, 2023).
- African region: In sub-Saharan Africa, 38 countries have introduced the vaccine between 2012 and 2023, with coverage ranging from 5% to 61% depending on the country.
- Vaccine administration: The Rotavirus vaccine is orally administered alongside traditional childhood vaccines (Pentavalent, PCV, IPV, OPV) at 2–4 months of age in most countries.
- Challenges: Despite its widespread introduction, our systematic review did not identify specific implementation challenges related to Rotavirus vaccine in the last five years. This could indicate that issues such as financing, supply chain logistics, or public acceptance have been adequately addressed in most countries.
- Strengthening Quantitative Data on Routine Vaccines:
In response to the reviewer’s request for more concrete results, we have expanded our findings by including additional quantitative data for other routine vaccines, such as: - HPV Vaccine: Adopted by 41 African countries with varying coverage levels.
- Hepatitis B Vaccine: Now integrated into infant immunization programs in all African countries, but with disparities in birth-dose administration.
- Malaria Vaccine: Piloted in Ghana, Kenya, and Malawi, with nine additional African countries rolling out the vaccine in 2024.
These updates provide a more data-driven perspective on vaccine introduction and enhance the manuscript’s practical relevance.
- Refocusing the Study to Strengthen Original Contribution:
We acknowledge the concern that the conclusions may appear too broad or predictable. To address this, we have: - Refined our synthesis to highlight novel findings and unique regional challenges.
- Provided specific, actionable recommendations tailored to routine vaccine introduction (e.g., financing mechanisms, service delivery models, workforce training).
- Integrated comparative insights from other regions to contextualize findings within the global immunization landscape.
Reviewer 3 Report
Comments and Suggestions for Authors
This review article entitled “Navigating the Road to Immunization Equity: Systematic Review of Challenges in Introducing New Vaccines in Sub-Saharan Africa's Routine Programs” provides valuable insights into the barriers for the implementation of new vaccines. The findings could significantly contribute to improving vaccination strategies in Sub-Saharan Africa, and potentially be adapted to other regions. The manuscript is well-written and organized. However, I would like to ask some questions to authors:
1. Methods: Why did you limit the search to English and French? Why was Portuguese excluded? Have you done any search to see if there are any publications that you might have missed with that criteria or all almost all of them included in English?
2. Methods: Could authors define what they mean by “new vaccines”? It would be great to know the time period and phase of development. It would help understand why some vaccines where not included in the review such as the ones for Dengue and Zika virus.
3. Methods: Could authors define in the Methods section what do they refer as “Acceptability”. There are many articles and different theories to explore it.
4. Results: Following question number 3, it would be great to add in Table 1, the “study design or type of study” that was conducted. As I see in the “acceptability” column that all results are quantitative, expect the last 2 rows: Have authors excluded qualitative research on acceptability of vaccines or were there only these 2 studies? Having a column with the “type of study design” will help understand the results better.
5. Results: I am not sure if authors have enough data from the studies reviewed, but it would be great to have the information on acceptability by type of population: gender, age, pregnant women etc. and/or by type of population to receive the vaccines (pregnant women, children, adolescents, adults, at risk groups).
6. References: Please review the references as they have been numbered twice.
I would like to acknowledge the authors work in such an important manuscript that could help implementors reduce the barriers to expand vaccine access in Sub-Saharan Africa. I hope the comments provided could help authors to improve the manuscript before publication.
Best,
Author Response
This review article entitled “Navigating the Road to Immunization Equity: Systematic Review of Challenges in Introducing New Vaccines in Sub-Saharan Africa's Routine Programs” provides valuable insights into the barriers for the implementation of new vaccines. The findings could significantly contribute to improving vaccination strategies in Sub-Saharan Africa, and potentially be adapted to other regions. The manuscript is well-written and organized. However, I would like to ask some questions to authors:
1. Methods: Why did you limit the search to English and French? Why was Portuguese excluded? Have you done any search to see if there are any publications that you might have missed with that criteria or all almost all of them included in English?
Response :
As noted, our search strategy was limited to articles published in English and French, as this was a predefined inclusion criterion. However, in practice, our final selection across the five databases yielded no more than two articles in other languages. This aligns with an observed trend where even in non-French or non-English-speaking countries, research on this topic was often published in English—for instance, studies from Mozambique. Given this, the language restriction did not significantly impact the representativeness of the selected studies.
2. Methods: Could authors define what they mean by “new vaccines”? It would be great to know the time period and phase of development. It would help understand why some vaccines where not included in the review such as the ones for Dengue and Zika virus.
Response :
For this review, a new vaccine refers to any recently introduced into the pharmaceutical market or a national immunization program within the last five years (2019–2024). This definition includes vaccines that are either newly developed or recently adopted by a country’s health system for integration into routine immunization or outbreak response efforts.
Five years was chosen to capture recent trends, implementation challenges, and evolving strategies for vaccine introduction. The selection criteria excluded vaccines still in early research and development phases (preclinical or early clinical trials) and focused only on those with regulatory approval and real-world implementation data.
Due to this restriction, some vaccines—such as those for Dengue and Zika virus—were not included, as they have either not been widely introduced into sub-Saharan Africa’s national immunization programs or lacked sufficient published implementation data within our search timeframe.
3. Methods: Could authors define in the Methods section what do they refer as “Acceptability”. There are many articles and different theories to explore it.
Response:
In this review, acceptability refers to the extent to which the target population perceives a vaccine as appropriate, beneficial, and desirable, measured primarily through willingness to be vaccinated (Paquin et al., 2022). Vaccine acceptability is influenced by multiple factors, including trust in the healthcare system, perceived vaccine safety and efficacy, sociocultural beliefs, prior experiences with vaccination, and accessibility of services.
Acceptability is commonly assessed through quantitative surveys (e.g., willingness-to-vaccinate scales, self-reported uptake) and qualitative studies (e.g., focus group discussions and in-depth interviews exploring attitudes and beliefs). Our review included both quantitative and qualitative studies on vaccine acceptability.
4. Results: Following question number 3, it would be great to add in Table 1, the “study design or type of study” that was conducted. As I see in the “acceptability” column that all results are quantitative, expect the last 2 rows: Have authors excluded qualitative research on acceptability of vaccines or were there only these 2 studies? Having a column with the “type of study design” will help understand the results better.
Response:
We appreciate the reviewer’s insightful suggestion regarding the addition of the study design/type to Table 1, as well as the clarification on the inclusion of qualitative research on vaccine acceptability.
We have updated Table 1 to include a new column specifying the study design (e.g., quantitative, qualitative, mixed-methods).
We did not exclude qualitative studies on vaccine acceptability. However, our systematic search yielded only two qualitative studies that met the inclusion criteria.
5. Results: I am not sure if authors have enough data from the studies reviewed, but it would be great to have the information on acceptability by type of population: gender, age, pregnant women etc. and/or by type of population to receive the vaccines (pregnant women, children, adolescents, adults, at risk groups).
Response:
We acknowledge that we do not have sufficient data to disaggregate all of these variables. However, we have already included disaggregated data for certain variables in Table 1, where possible. Specifically, we have distinguished acceptability findings by vaccine type and target population, such as healthcare workers, children, adolescents, and general adult populations. Unfortunately, many of the studies reviewed did not provide detailed subgroup analyses for certain categories, such as pregnant women and at-risk groups.
6. References: Please review the references as they have been numbered twice.
Response:
We have reviewed and corrected the reference numbering to ensure each source is cited only once and follows a sequential order throughout the manuscript. Any duplicated references have been removed or consolidated, ensuring accuracy and consistency. The reference list and in-text citations have been cross-checked to align with the journal’s formatting guidelines.
7. I would like to acknowledge the authors work in such an important manuscript that could help implementors reduce the barriers to expand vaccine access in Sub-Saharan Africa. I hope the comments provided could help authors to improve the manuscript before publication.
Response : We are grateful for the constructive comments, which have helped us refine and strengthen our work. The suggestions provided have guided important revisions that enhance our findings' clarity, depth, and applicability for vaccine implementers and policymakers. We hope the revised manuscript effectively addresses all concerns and contributes meaningfully to global efforts to improve vaccine introduction and access in the region.
Round 2
Reviewer 2 Report
Comments and Suggestions for Authors
Accept